# Prenatal Mercury Exposure in Pregnant Women from Suriname’s Interior and Its Effects on Birth Outcomes

**DOI:** 10.3390/ijerph17114032

**Published:** 2020-06-05

**Authors:** Gaitree K. Baldewsingh, Jeffrey K. Wickliffe, Edward D. van Eer, Arti Shankar, Ashna D. Hindori-Mohangoo, Emily W. Harville, Hannah H. Covert, Lizheng Shi, Maureen Y. Lichtveld, Wilco C.W.R. Zijlmans

**Affiliations:** 1Medical Mission Primary Health Care Suriname, Paramaribo, Suriname; evaneer@gmail.com; 2Faculty of Medical Sciences, Anton de Kom University of Suriname, Paramaribo, Suriname; wilco.zijlmans@uvs.edu; 3School of Public Health and Tropical Medicine, Tulane University, New Orleans, LA 70112, USA; jwicklif@tulane.edu (J.K.W.); sarti@tulane.edu (A.S.); ashna.mohangoo@perisur.org (A.D.H.-M.); eharvill@tulane.edu (E.W.H.); hcovert@tulane.edu (H.H.C.); lshi1@tulane.edu (L.S.); mlichtve@tulane.edu (M.Y.L.); 4Foundation for Perinatal Interventions and Research in Suriname (Perisur), Paramaribo, Suriname; 5Scientific Research Center Suriname/Academic Hospital Paramaribo, Paramaribo, Suriname

**Keywords:** mercury, prenatal exposure, birth outcomes, indigenous, tribal, Suriname

## Abstract

Prenatal mercury (Hg) exposure was determined in a sub-cohort of the Caribbean Consortium for Environmental and Occupational Health’s environmental epidemiologic prospective cohort study of pregnant women living in Suriname’s interior. The associations between Hg exposure, low birth weight (LBW, <2500 g) and preterm birth (PTB, <37 weeks) were explored. Correlation analysis, Fisher’s exact test and logistic regression analyses were conducted to evaluate the associations between maternal hair Hg levels and birth weight, LBW and PTB, and between potential confounders, LBW and PTB, respectively. Among 204 singleton births were 198 live births, five stillbirths and one miscarriage. The mean participant age was 26 years; 15.7% of participants had PTBs and 8.1% delivered a child with a LBW. The median hair Hg level was 3.48 μg/g hair. Low hair Hg exposure, based on lowest tertile < 2.34 μg/g, was associated with LBW (OR = 7.2; 95% CI 1.5–35.6; *p* = 0.015); this association was independent of maternal age, ethnic background, household income and village location, and no correlation was found between hair Hg and PTB. Young maternal age was associated with PTB (RR = 5.09, 95% CI: 1.92–13.85; *p* = 0.0004) while maternal age was not associated with hair Hg or LBW. The impact of prenatal Hg exposure on pediatric neurodevelopment is currently being evaluated in the infant sub-cohort.

## 1. Introduction

Mercury (Hg) is an environmental pollutant with high toxic potential. The fetus and the developing child are particularly sensitive to its harmful effects. The most hazardous form for human health is methylmercury, which is formed when the inorganic Hg that is discharged from various contaminants in the environment flows into rivers and streams. It then is converted by bacteria in water and accumulates in food chains consisting mainly of fish [1]. Hg can readily cross the placenta as methylmercury and potentially affect the fetus [1,2]. Due to the potential toxicity of Hg to the fetus, fish intake during pregnancy has been of special interest worldwide [3,4]. For people living in Suriname’s interior, where Hg is widely used in artisanal gold mining activities, the main food source for protein is fish [5].

Suriname is a middle-income country, situated on the north-east coast of South America, with an estimated population of 575,991 [6]. Approximately 10% of the population lives in the remote tropical rainforest interior. The majority of this population (83%) identifies as tribal (descendants of runaway African slaves) and 17% are of indigenous descent (native Amerindians). These populations live in relatively small communities with their own cultural beliefs and many speak their own native language. The communities are mostly situated far from the capital in remote areas near large rivers and are only accessible by air or water. Their diets are mostly dependent on locally caught fish or hunted meat. The Medical Mission Primary Health Care Suriname (MMPHCS) is the primary and sole healthcare provider in the interior and has introduced the basic principles of primary healthcare according to the Alma Ata declaration: equitable distribution of healthcare, community participation, health workforce development, use of appropriate technology and a multi-sectoral approach [7]. MMPHCS provides antenatal care (ANC) as part of its maternal and childcare portfolio and follows the WHO’s recommendations to improve maternal and neonatal health outcomes.

Low birth weight (LBW) and preterm birth (PTB) are important predictors of under-five mortality. Globally, 15.5% of all newborns are born with LBW, of which 96% are born in developing countries [8]. In Suriname, the under-five mortality rate is 18.9% [9]. The prevalence of LBW in Suriname is 14.7% [10] and according to the Global Burden of Disease study that was conducted in 2010 (GBD), complications resulting from premature birth are the third largest cause of perinatal mortality and disability in Suriname [11]. The interior of Suriname contributes to approximately 10% of the live births nationwide [12]. Hg exposure in pregnancy has been associated with both complications during pregnancy and developmental problems in infants [11]. A negative association between Hg exposure and birth size [12], and between fetal Hg exposure and growth restriction, has been reported repeatedly [2,13,14]. In Spain, a two-fold increase in cord blood total Hg was associated with a significant reduction in birth weight and head circumference at birth, an important indicator of brain development [14]. In a Michigan study, women who delivered before 35 weeks of gestation more often had hair Hg levels above the 90th percentile (≥0.55 µg/g hair), even after adjusting for mothers’ characteristics and fish consumption [15]. Information is lacking on the extent to which pregnant women living in the interior of Suriname are exposed to Hg and the extent to which this exposure contributes to adverse birth outcomes such as LBW and PTB.

The Caribbean Consortium for Research in Environmental and Occupational Health’s (CCREOH) MeKiTamara environmental epidemiology cohort study assesses prenatal exposure to non-chemical and chemical stressors, including heavy metals such as Hg, and their association with increased adverse birth and pediatric health outcomes in Suriname. Surinamese pregnant women were recruited in three different regions: in the capital Paramaribo, in Nickerie as the largest agricultural district and in the interior. Due to the active artisanal and small-scale goldmining operations in the interior, we expected Hg exposure to be higher in the interior compared to the other two regions. This study aimed to determine prenatal Hg exposure in the subset of pregnant women living in Suriname’s interior and to explore the potential association between Hg exposure and the adverse birth outcomes LBW and PTB. We hypothesized that Hg exposure in these women would be positively correlated with the adverse birth outcomes.

## 2. Materials and Methods

### 2.1. Data Sources

In this prospective cohort study, pregnant women were recruited from April 2017–December 2018 at 15 randomly selected MMPHCS health centers as part of the CCREOH MeKiTamara cohort study. Both tribal and indigenous communities were sampled. The council of village heads was approached, informed about the study and asked to assign the villages eligible for recruitment. A total of 206 pregnant women was recruited, all of whom had registered at one of the 15 MMPHCS health centers in these villages. Six of the 15 interior villages included were easily accessible by road from the capital Paramaribo, situated in the rural urban region of Suriname, while the remainder were reachable only by boat and air. Only singleton pregnancies were included in the study. Written informed consent in participants’ native languages—Saramaccan for the tribal and Trio for the indigenous participants—was obtained. In the cultural setting of the Surinamese interior, persons of 16 years and older are considered adults; therefore, women from 16–45 years of age were eligible for recruitment. Ethical approval was granted by the Institutional Review Board of the Ministry of Health of Suriname (VG 023-14) and the Institutional Review Board of Tulane University, New Orleans, LA. All the women gave written informed consent and, if applicable, informed assent.

### 2.2. Protocols

All pregnant women were asked to provide a hair sample, ideally before 27 weeks of gestation. Hair samples of at least 1.5 g were collected according to protocol by cutting strands close to the scalp from the occipital region of the head, stored at room temperature in a climate-controlled room and sent to the Environmental Research Center lab of the Anton de Kom University of Suriname for total Hg analysis using cold vapor atomic absorption spectrometry (CVAAS). A total number of 178 hair samples was analyzed. Socio-demographic variables were re-categorized for the logistic models as follows: age at delivery (<20 vs. 20–34 vs. ≥35 years); parity nulliparous or one previous live birth vs. 1–4 previous live births vs. ≥5 previous live births; ethnic background (“tribal community” vs. “native community” (indigenous)); and location of the village (downstream vs. upstream vs. not near the goldmining activities). Data on birth weight and gestational age were collected prospectively and adverse birth outcomes were defined as LBW (birth weight <2500 vs. 2500+ g) and PTB (<37 vs. 37+ completed weeks of gestation). Gestational age was calculated based on the last menstruation period. All LBW were included in statistical analyses regardless of whether PTB was also identified for that same participant. The US Environmental Protection Agency (USEPA) reference dose for methylmercury is 0.1 μg/kg body weight/day, which corresponds to a hair mercury concentration of 1.1 μg/g [16]. The USEPA uses 1.1 μg/g as the health action threshold for decisions regarding intervention or follow-up. 

### 2.3. Statistical Analysis

Hair Hg level in the study population was analyzed with the one sample t-test for significance.

#### 2.3.1. Mercury and Birth Outcomes

##### Contingency Table Analyses

We dichotomized the hair Hg data using the median hair Hg concentration of 3.48 μg/g. Participants were categorized as either high (above the median) or low (below the median) exposure groups. In addition, tertiles of hair Hg concentration were used to examine any crude trends using the chi-square test, with participants divided into the low (the lowest tertile, <2.34 µg/g), medium (the middle tertile, 2.34–6.99 µg/g) and high (the highest tertile, >6.99 µg/g) exposure groups.

##### Regression Analyses

Linear (with infant birth weight (in grams) as a continuous variable) and logistic (with PTB and LBW as binary variables) regression analyses were carried out, also adjusting for maternal age, ethnic background, household income and village location. A segmental linear regression was carried out, setting X0 = 3.48, in order to explore the differences in linearity below and above the median hair Hg concentration.

#### 2.3.2. Maternal Age and Birth Outcomes

##### Contingency Table Analyses

We dichotomized age in years at the mean (26 years) as well as in three subgroups to examine any crude trends, using the chi-square to test the differences across the lowest subgroup (<20 years), middle subgroup (20–34 years) and oldest subgroup (≥35 years).

##### Regression Analyses

Linear and logistic regression analyses were carried out, treating infant birth weight (in grams) as a continuous variable for linear regression analysis and coding PTB and LBW as binary variables for simple logistic regression.

#### 2.3.3. Gender Comparison among Newborns

Newborn infant gender and birth outcomes (PTB and LBW) were analyzed using Fisher’s exact test. We also compared maternal hair Hg and gender using the Mann–Whitney U test.

All analyses were conducted using Prism ver. 8.3.0 (GraphPad; San Diego, CA, USA) and the Statistical Package for Social Science version 24. *p*-values < 0.05 were considered significant.

## 3. Results

A total of 204 women (99%) from tribal and indigenous communities ultimately participated in this study. All had registered at MMPHCS clinics for ANC. One woman was excluded because of twin pregnancy and one moved abroad. The 178 participating women provided suitable hair samples. One woman had a miscarriage. In total, 203 women delivered infants, with 198 live births and five stillbirths. Of the 198 live births, 96 (48%) were from tribal and 102 (52%) from indigenous women (Figure 1). Four children were born with congenital anomalies (one with Down syndrome, three with hydrocephalus). One of the five stillbirths had congenital anomalies (not specified).

Participants’ characteristics are shown in Table 1. The overall mean age of the 204 included women was 26.3 (SD 7.26) years. Of these women, 22.5% were younger than 20 years, of which 30 (65.2%) were from the indigenous community, and 32 (15.7%) women delivered preterm. The infant mean birth weight was 3064 g (SD 483). The overall rates of adverse birth outcomes were 8.1% for LBW and 15.7% for PTB. 

Figure 2 shows the hair Hg concentration levels in the study population. Of the women in the interior, 93% had significantly elevated hair Hg levels that well exceeded the USEPA action level of 1.1 μg/g hair (t = 12.595, *p* < 0.001). The distribution of mercury exposure in the three regions within Suriname where CCREOH participants were recruited was almost non-overlapping and was substantially higher in the interior sub-cohort compared to the other two regions: median total Hg concentrations in hair from pregnant women in the interior (*n* = 178) were 3.48 µg/g hair (interquartile ranges (IQR) 2.01–9.01; range 0.38–31.91), from the capital Paramaribo (*n* = 482) 0.63 µg/g (IQR 0.36–1.09; range 0.00–11.88) and from the Nickerie district (*n* = 188) 0.77 µg/g (IQR 0.50–1.05; 0.00–14.01).

Figure 3 below shows the correlation between participants’ age and hair Hg concentrations. There was no significant correlation between participants’ age and hair Hg concentration (Spearman r = 0.10, 95%CI −0.25–0.05, *p* = 0.16)

### 3.1. Mercury and Birth Outcome Analyses

There was no significant association between PTB and hair Hg concentration in mothers during pregnancy. The relative risk of PTB and high hair mercury concentration was 1.55 (95% CI 0.75–3.23, *p* = 0.29). The relative risk of LBW and high hair mercury concentration was 0.52 (95% CI 0.17–1.55, *p* = 0.38). No significant trends were noted for either PTB (χ^2^ = 0.09, *p* = 0.76) or LBW (χ^2^ = 2.10, *p* = 0.14) with increasing concentrations of mercury in hair. Since most Hg levels well exceeded the USEPA action level of 1.1 µg/g hair (t = 12.595, *p* < 0.001), we reanalyzed the data with the USEPA cut-off of 1.1 µg/g and found no differences between the mothers with hair Hg levels below 1.1 µg/g and mothers with hair Hg levels ≥ 1.1 µg/g for LBW (*p* = 0.276) or PTB (*p* = 0.647).

#### 3.1.1. Regression Analyses

The linear regression of birth weight against maternal hair Hg concentration did not support a significant relationship (Figure 4). The estimated slope of 10.26 (95% CI −4.54–25.05, r^2^ = 0.01) was not significantly different from 0 (*p* = 0.18). The crude logistic regression did not indicate an association between above vs. below the median hair Hg levels with regard to either PTB (19.5% vs. 10.5%; *p* = 0.132) or LBW (8.3% vs. 11.8%; *p* = 0.460). The segmental linear regression indicated that there was no relationship between maternal hair Hg concentration and birth weight either below the median or above the median of 3.48 µg/g (r^2^ = 0.015; lower slope = 76.7 95% CI −24.3–177.7, upper slope = 1.6 95% CI −18.1–21.3). When comparing women with hair Hg levels in the lowest tertile group (<2.34 μg/g) vs. those in the middle and high tertile groups, the proportion of LBW babies was significantly higher among participants with low hair Hg exposure compared with participants with medium/high hair Hg exposure (18.4% vs. 6.3%; *p* = 0.019). The crude logistic regression revealed a 3.34 increased risk of LBW (95% CI: 1.17–9.58, *p* = 0.025). The multivariate logistic regression revealed that participants with a low Hg exposure had significantly higher odds of LBW compared with participants with medium/high hair Hg exposure (OR = 7.2; 95% CI 1.5–35.6; *p* = 0.015); this association was independent of their age, ethnic background, household income and village location.

#### 3.1.2. Mercury and Gender

The assessment of maternal hair Hg concentration and infant gender showed no significant difference in maternal hair Hg concentrations between female and male children (Mann–Whitney U = 3737, *p* = 0.83).

### 3.2. Maternal Age and Birth Outcome Analysis

There was a significant association between younger maternal age and PTB. The relative risk of PTB in participants with a younger maternal age (<25 years) was 5.09 (95% CI 1.92–13.85, *p* = 0.0004). There was no significant association between younger maternal age and LBW. The relative risk of LBW in participants with a younger maternal age was 1.36 (95% CI 0.47–3.94, *p* = 0.76). A significant trend was noted for PTB and decreasing maternal age (χ^2^ = 14.79, *p* = 0.0001), but no trend was noted for LBW (χ^2^ = 0.36, *p* = 0.55) and maternal age. We compared the two high risk groups, <20 years and ≥35, with the age group 20–34 years for PTB and LBW. There was no significant trend seen for either PTB (χ^2^ = 2.88, *p* = 0.89) or LBW (*p* = 0.763, Fisher’s exact test), respectively.

#### Regression Analyses

The linear regression of birth weight against maternal age suggested a small, positive relationship (Figure 5). The estimated slope of 9.98 (95% CI 0.64–19.31, r^2^ = 0.02) was significantly different from 0 (*p* < 0.04). 

The crude logistic regression analysis indicated that there was an association between maternal age and PTB (*p* < 0.0003), with the probability of PTB increasing as maternal age decreased, but not for maternal age and LBW (*p* < 0.999).

### 3.3. Gender and Birth Outcome Analysis

Overall, there were more male infants (59.9%) born than female (40.1%). There was a higher incidence of PTB among male compared with female newborns (17.4% vs. 8.2%), although this result was not significant (*p* = 0.077). 

LBW was more often seen in female than in male newborns (12.5% vs. 8.3%), but the observed difference was not statistically significant (*p* = 0.350). 

There was no difference in maternal age between female and male children (Mann–Whitney U = 4470, *p* = 0.71).

## 4. Discussion

Our study focused on the prenatal exposure to Hg of pregnant women living in the interior of Suriname and the prevalence of the adverse birth outcomes LBW and PTB in their offspring. Hair Hg levels in 93% of these women were well above the USEPA action levels. Low Hg exposure in these women was associated with LBW; our results do not suggest an association between mercury exposure and the occurrence of PTB in these women. While younger women more often delivered prematurely, we did not find an association between maternal age and mercury exposure or low birth weight. There were no gender differences in the adverse birth outcomes.

The hair Hg levels in our study population were three times higher than the USEPA health action threshold. When we split the levels at the USEPA cut-off of 1.1 µg/g for the analysis, we found no differences between mothers with hair Hg levels below and above the cut-off point for LBW or PTB. We did find women with low hair Hg levels to have a greater risk of delivering a child with LBW. People living in the interior of Suriname heavily depend on locally caught fish for their daily protein intake. High concentrations of Hg have been found in fish collected from areas where artisanal and small-scale goldmining occurs [17]. Most probably, the vast majority of pregnant women in this study were primarily exposed to Hg from consuming contaminated fish. Fish that have high levels of Hg may also be rich in selenium and/or polyunsaturated omega-3 fatty acids (PUFA-3), which are known to have beneficial nutritional and possibly neuroprotective characteristics which could potentially outweigh the neurotoxic effects of Hg [18]. Hg levels in fish generally increase with the increasing length, weight and age of the fish [19,20]; this pattern likely reflects the bio-concentration of Hg in the food chain. Since people from the interior eat more carnivorous fish that tend to have higher Hg levels, the corresponding selenium and PUFA-3 levels should be assessed in these fish. The lack of association between high maternal hair Hg concentrations and adverse birth outcomes could well be influenced by the potential protective roles of selenium and PUFA-3. 

Birth weight was related to Hg exposure in this study: women with low Hg exposure had a higher chance of delivering a newborn with LBW compared to women with medium or high exposure. This association has been described previously, where it was suggested that the protective effects of high fish consumption counteract Hg exposure [21,22,23,24]. Methylmercury (MeHg) has the potential to cross the placenta and negatively affect the developing fetus [25]. Studies on hair mercury levels as related to fetal growth are contradictory. In a longitudinal study in Seychelles, prenatal MeHg exposure (mean 3.92 ppm) and maternal long chain polyunsaturated fatty acids were not associated with birth weight [26]. In a Spanish cohort (*n* = 554), newborns with cord blood mercury in the highest quartile weighed 143.7 g less (95% confidence interval (CI): −2251.8, −235.6) than those in the lowest quartile, after adjusting for fish consumption and other variables [27]. The amount of fish consumption and the type of fish consumed during pregnancy may have an important role in fetal growth. Fish consumption advice during pregnancy is designed to be protective of infant growth and neurodevelopment by reducing exposure to mercury. In addition to being a healthy source of dietary protein, recent research indicates that additional aspects of fish consumption may offer neuroprotective benefits to developing children including omega-3 fatty acids (PUFA-3) and selenium (Se) [21,22,23,24]. The prevalence of LBW in our interior study population (8.1%) is comparable to the country-wide estimate (9.3% [28]), as well as the US (8.3% [29]) and Venezuela (8.6%) [30], and it is lower than in Brazil (12.3%) [30]. We found no differences in LBW between the tribal and indigenous women included in our study.

Despite the fact that the vast majority of our participants had experienced significant exposure to Hg, we did not find an association with PTB. Our findings are consistent with those found by some others. No association was observed between gestational age and cord blood Hg in a Canadian study [31], a New York City cohort with maternal or cord blood Hg [32] or a French cohort with maternal hair [33]. In a Michigan study, however, women who delivered very preterm (<35 weeks) were more likely to have high hair Hg levels (0.55–2.5 µg/g) than women who delivered at term [15].

Even though interior women had higher hair Hg concentrations compared to the coastal area, the incidence of adverse birth outcomes was lower than the overall CCREOH cohort (LBW 8.1 vs. 14.7 and PTB 15.7 vs. 18.4, Zijlmans et al., cohort profile CCREOH MeKiTamara cohort study, unpublished results). These findings suggest that, in addition to environmental toxicants, several other (epidemiologic) factors may influence the prevalence of LBW and PTB, such as maternal socio-demographics, reproductive history and medical conditions. The lower prevalence of adverse birth outcomes in the interior population may also in part be a result of the consumption of fatty acids and other essential nutrients that potentially counter the toxic effects of Hg present in locally harvested fish. Nevertheless, in the interior, few alternative sources of protein are available, which remains of concern. In addition, although we found limited associations between hair Hg and adverse birth outcomes, it is critically important to assess the long-term effects of Hg exposure on the children in this cohort, particularly their neurocognitive development. In this respect, the neurodevelopmental testing of these children is currently being conducted and analyses of these results are underway.

There was a higher incidence of PTB in male newborns, whereas LBW was more often seen in female newborns. These findings on the association between gender and adverse birth outcomes have been repeatedly reported and are commonly accepted as standard features of birth/gender epidemiology [34,35,36]. In addition, maternal Hg exposure and the gender of the fetus were not associated, making gender unlikely to be a confounder.

Maternal age and Hg exposure were not associated in our study. Santos et al. observed, in a study conducted in the Brazilian Amazon, that maternal age was positively associated with Hg exposure. Similar findings were reported from a 1999–2000 National Health And Nutrition Examination Survey (NHANES) in which hair Hg levels were assessed in women of childbearing age (16–49 years) and children from 1 to 5 years old [37]. We found a positive association between maternal age and PTB and younger mothers had a higher risk of delivering prematurely. Similarly, mothers who had one or less parities had a higher chance of PTB compared to mothers with multiple parities. The present study found no significant relationship between maternal age and LBW. 

Continuation of the MMPHCS standardized maternal services in the communities within remote areas is pivotal to emphasizing the risk of Hg exposure to women of reproductive age, particularly those of adolescent age. Introducing targeted health education programs should ensure that pregnant women are offered dietary alternatives; for example, advisories directed towards the consumption of non-predatory fish species that are less contaminated with Hg and are rich in potential protective nutrients. Besides the information provided through MMPHCS healthcare workers, reaching out to the women through the traditional village head or even the traditional birth attendants could significantly improve their knowledge and insight into this matter. Public education on the risk of Hg exposure to pregnant women and developing children should be strengthened in order to change the communities’ perception of exposure to neurotoxic substances.

To our knowledge, the present study is the first to examine prenatal exposure to Hg and birth outcomes in women living in remote areas in the interior of Suriname. Newborn data was collected directly, both from hospital medical records and from the interior primary healthcare facilities. The MMPHCS health facilities have a standardized health care system with early detection and early referral of women with pregnancy-related risk factors. Moreover, participant adherence in this study was almost 100% which may be attributed to a well-functioning MMPHCS healthcare system. The use of community-based healthcare facility data allows us to identify tailored mother and child interventions as part of our monitoring of the prenatal services provided.

We note that there were some limitations regarding the designation of PTB. Data on PTB were estimated from the last menstrual period which was reported by each participant as ultrasound was not available. This may have introduced recall bias, leading to an over- or underestimation of PTB incidence. Additionally, not all of the known risk factors were considered, including maternal medical conditions and the history of LBW and PTB. 

## 5. Conclusions

The vast majority of women living in the interior of Suriname had hair mercury levels well above the international accepted action levels. While Hg exposure in interior pregnant women is of significant concern, we found low Hg exposure to be associated with LBW, suggesting the protective effects of the high consumption of fish containing beneficial nutrients that counteract Hg exposure. Young women more often delivered prematurely; maternal age was neither associated with low birth weight nor with mercury exposure. Our results are consistent with the literature regarding exposure to Hg during pregnancy at these concentrations. As there are several other epidemiologic factors influencing birth outcomes, the findings related to birth outcomes in this sub-cohort will be validated by ongoing analyses of the larger CCREOH study cohort (*N* = 1069). Results relating to the fish consumption of these women are pending. A follow-up on the potential long-term effects of exposure to toxicant mixtures such as mercury and lead in infants and toddlers is critical and is currently being done through pediatric neurodevelopmental assessments.

The ongoing monitoring of mercury in both humans and fish is necessary to ensure the effectiveness of public health risk management. 

## Figures and Tables

**Figure 1 ijerph-17-04032-f001:**
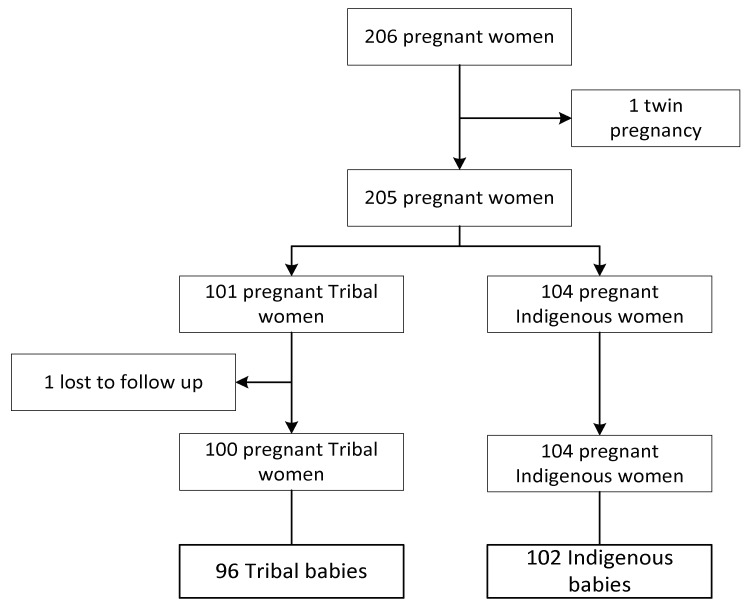
Enrolment flowchart.

**Figure 2 ijerph-17-04032-f002:**
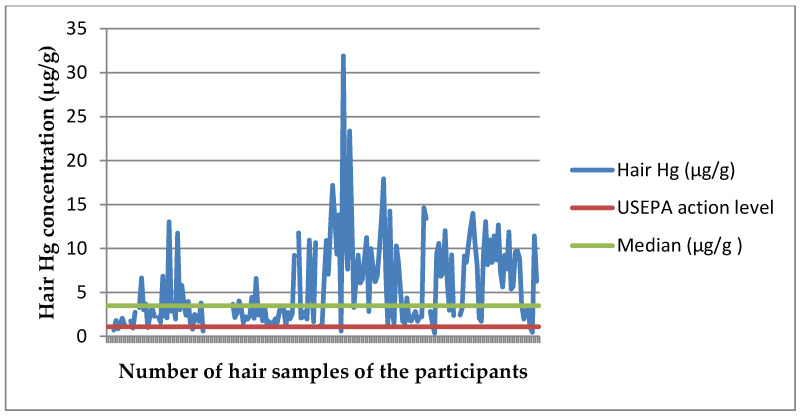
Total mercury concentrations in hair samples of 178 study participants.

**Figure 3 ijerph-17-04032-f003:**
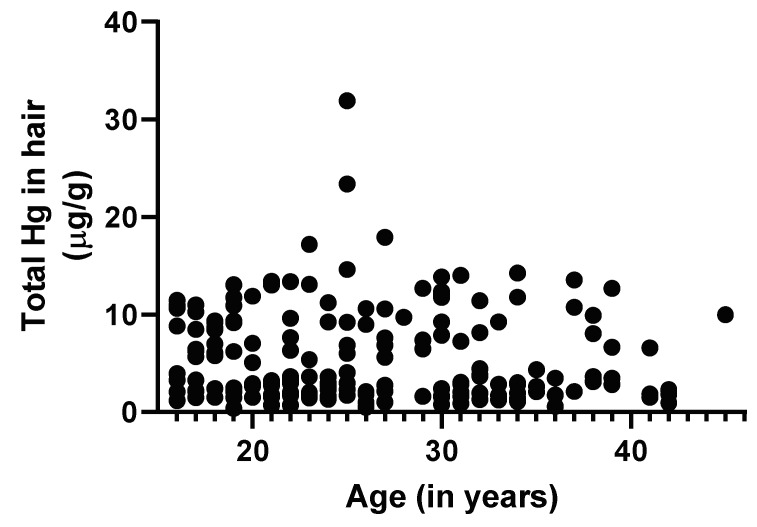
Scatterplot of participants’ age in years and concentrations of total mercury in hair. This included 178 participants. Total Hg concentrations in hair were not significantly correlated with age (r = −0.10, 95% CI −0.25–0.05, *p* = 0.16).

**Figure 4 ijerph-17-04032-f004:**
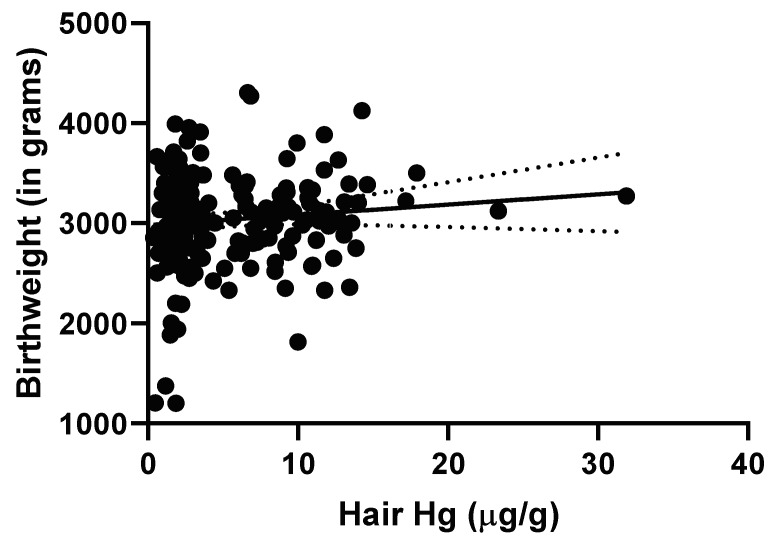
Linear regression of birth weight (in grams) against maternal hair Hg concentration. Estimated slope of the regression line is 10.26 (95% CI −4.54–25.05, r^2^ = 0.01, *p* = 0.18).

**Figure 5 ijerph-17-04032-f005:**
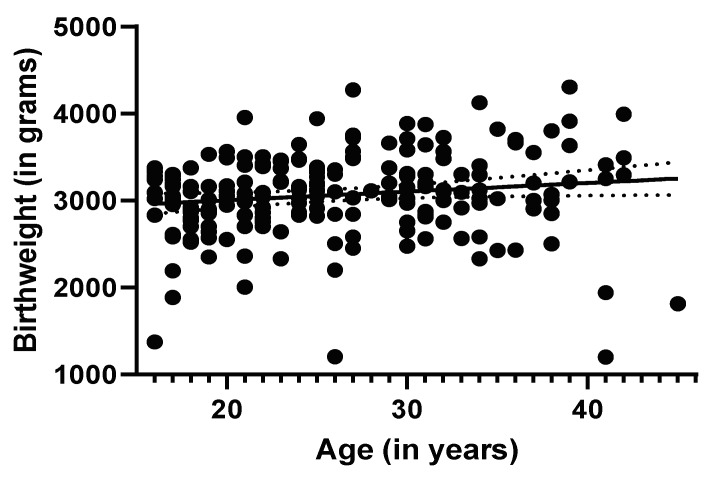
Positive relationship between maternal age (in years) and birth weight (in grams). Estimated slope of the regression line is 9.98 (95% CI 0.64–19.3, r^2^ = 0.02, *p* < 0.04).

**Table 1 ijerph-17-04032-t001:** Characteristics of 204 participants.

Maternal Characteristics	*N* (%)	Tribal (*n* = 100) *N* (%)	Indigenous (*n* = 104) *N* (%)
Age (years)			
16–19	46 (22.5)	16 (34.8)	30 (65.2)
20–34	125 (61.3)	63 (50.4)	62 (49.6)
≥35	33 (16.2)	21 (63.6)	12 (36.4)
Parity			
≤1	74 (36.3)	31 (41.9)	43 (58.1)
1–4	54 (26.5)	24 (44.4)	30 (55.6)
≥5	74 (36.3)	44 (59.5)	30 (40.5)
Missing	2 (1.0)	1 (50.0)	1 (50.0)
Gestational age (wks.)			
<37	32 (15.7)	11 (34.4)	21 (65.6)
37–40	148 (72.6)	77 (52.0)	71 (48.0)
≥41	22 (10.8)	12 (54.5)	10 (45.5)
Missing	2 (0.9)		2 (100)
Birth weight (grams)			
<2500	16 (7.8)	8 (50.0)	8 (50.0)
≥2500	182 (89.6)	88 (48.4)	94 (51.6)
Stillbirth	5 (2.5)	3 (66.7)	2 (33.3)
Miscarriage	1 (0.1)	1 (100.0)

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
