# Peer review of "Prenatal Mercury Exposure in Pregnant Women from Suriname’s Interior and Its Effects on Birth Outcomes"

_ijerph, 2020, doi:10.3390/ijerph17114032_

Round 1
Reviewer 1 Report
In this study the authors assessed the potential association of the exposure to Prenatal Hg and the adverse birth outcomes low birth weight (LBW, <2500g) and preterm birth (PTB, <37weeks) in a sub-cohort of the Caribbean Consortium of Environmental and Occupational Health environmental epidemiologic cohort study of pregnant women living in Suriname’s interior. Hg exposure in pregnancy has been associated with both complications during pregnancy and developmental problems in infants, however a negative association between Hg exposure and birth size and fetal growth have been reported repeatedly.
However, the Hg levels in hair in a large percentage (93%) of these women were found well above the USEPA action levels (3 times higher), most probably due to consuming of contaminated fish.
People living in the interior of Suriname strongly depend on local fish for their protein intake and high concentrations of Hg have been found in fish collected from areas where eventually ASGM is a wide phenomenon.
Despite this, results from this study do not suggest an association between mercury Hg and both PTB and LBW in these women. Younger women more often delivered prematurely, however authors do not find an association between maternal age and Hg exposure.
Authors suggest also that this low prevalence of adverse birth outcomes could be a result of the simultaneous fatty acids and other nutrients presence in locally harvested fish that potentially counter toxic effects of Hg.
These findings suggest that complex interactions between environmental Hg exposure and several other factors could influence the occurrence of both LBW and PTB, such as nutrient factors in diet, socio-demographics and medical conditions of these women.
I recommend this paper for the publication, because this study highlights how high levels of Hg do not affect negatively pregnancy outcomes of women belonging to population exposed to toxic effects of Hg, and further studies are needed to investigate the complex pattern between biology, chemistry, genetic and social and other conditions.
There are minor editorial issues in figures, like 3 and 4.
Author Response
Dear Reviewer,
Thank you for your insightful comments. We have tried to address them to the extent possible and have revised the manuscript accordingly.
There are minor editorial issues in figures, like 3 and 4.
Reply: thank you for pointing this out, language corrections were made throughout the manuscript. We corrected the font in the figures 3 and 4.
Also, in reanalyzing the data, we encountered some irregularities in our dataset leaving us 179 maternal hair samples instead of 184. Therefore adjustments were made as to the numbers of medians and tertiles.
Reviewer 2 Report
A strength of the study was to collect hair sample of pregnant women and measure the hair Hg level of pregnant women in Suriname. But the weekness of the study is the research design and analysis methods.
According to the USEPA reference, the cut-off is 1.1 ug/g, why not use the USEPA cut-off in the analysis?
According to Figute 4, it seems there is non-liner association between birthweights (in grams) and maternal hair Hg concentration, should try add the square term in the regression analysis.
As there are many risk factors for PTB or LBW, so it is better to adjust other confounders in the regression analysis, for example, maternal food consumption, the SES information and the village dummy, etc.
Is there any gender difference between girls and boys.
1,According to the manuscript, the sample pregnant women were recruited in three different regions: in the capital Paramaribo, in Nickerie as the largest agricultural district, and in the interior. But this study only focused on pregnant women living in Suriname's interior. Need to argue why only include the pregnant women living in Suriname's interior in the analysis. I think it will be more interesting to use all sample in the analysis.
2,According to the manuscript, the median hair Hg concentration (3.29 μg/g) is about three times of the USEPA threshold (1.1 μg/g) . If there is a jump of birth weight (LWB or PTB ) at the USEPA threshold (1.1 μg/g), how to identify the jump effect at the USEPA threshold if separated all samples into tertiles. So need to provide evidence why separated all samples into tertiles in the analysis. I think it will be better to use the USEPA cut-off in the analysis as a robustness check.
3,As there are many risk factors for PTB or LBW, so it is better to adjust other confounders in the regression analysis if possible, for example, food consumption (especially the fish consumption) of pregnant women, the SES information of sample family, and the village dummy, etc.
4,If the fish consumption of pregnant women is available, it will be interesting to test the mediation effect of Hg concentration of pregnant woman on the association between fish consumption of pregnant women and birth weight (LWB or PTB ).
5, As gender is an importnant issue, it will be interesting to report the difference between girls and boys in the analysis.
Author Response
Dear Reviewer,
Thank you for your valuable comments, we have addressed these to the extent possible and hope that this complies with your remarks.
A strength of the study was to collect hair sample of pregnant women and measure the hair Hg level of pregnant women in Suriname. But the weakness of the study is the research design and analysis methods.
Reply: thank you for your valuable comments, we have addressed these to the extent possible and hope that this complies with your remarks.
In reanalyzing the data, we encountered some irregularities in our dataset leaving us 179 maternal hair samples instead of 184. Therefore adjustments were also made as to the numbers of medians and tertiles highlighted in the text.
According to the USEPA reference, the cut-off is 1.1 ug/g, why not use the USEPA cut-off in the analysis?
Reply: thank you for this suggestion. We reanalyzed the data with the USEPA cut-off of 1.1 µg/g, we found no differences between the mothers with hair Hg levels below 1.1 µg/g and mothers with hair Hg levels ≥ 1.1 µg/g for LBW (p=0.276) or PTB (p=0.647). This has been added to the results section lines 230-236 and discussion section lines 330-332.
According to Figure 4, it seems there is non-liner association between birthweights (in grams) and maternal hair Hg concentration, should try add the square term in the regression analysis.
Reply: we saw no clear nonlinearity or curvilinearity but instead a random scatter plot with no relationship between the variables. To verify this we performed a segmented linear regression for birthweight and maternal hair [Hg]. As before, we found no relationship between these to variables (r2=0.015). The lower and upper slopes (split on the median) were not different from zero. We have added this in result section lines 257-259.
As there are many risk factors for PTB or LBW, so it is better to adjust other confounders in the regression analysis, for example, maternal food consumption, the SES information and the village dummy, etc.
Reply: Please see below in remark no. 3
Is there any gender difference between girls and boys.
Reply: Please see below added in remark no.5
1,According to the manuscript, the sample pregnant women were recruited in three different regions: in the capital Paramaribo, in Nickerie as the largest agricultural district, and in the interior. But this study only focused on pregnant women living in Suriname's interior. Need to argue why only include the pregnant women living in Suriname's interior in the analysis. I think it will be more interesting to use all sample in the analysis.
Reply: thank you for this comment, pregnant women were indeed recruited from three different regions. The distribution of mercury exposure in the three regions was almost non-overlapping – substantially higher in the interior compared to the other two regions. For this reason, we focused on the interior region separately in this paper. Also the health system serving the population living in the interior as well as the dietary habits of its inhabitants including the interior participants in this study are different than in the capital and coastal area.
2,According to the manuscript, the median hair Hg concentration (3.29 μg/g) is about three times of the USEPA threshold (1.1 μg/g) . If there is a jump of birth weight (LWB or PTB) at the USEPA threshold (1.1 μg/g), how to identify the jump effect at the USEPA threshold if separated all samples into tertiles. So need to provide evidence why separated all samples into tertiles in the analysis. I think it will be better to use the USEPA cut-off in the analysis as a robustness check.
Reply: thank you for this comment, please see our earlier reply. No differences between the 2 groups for LBW and PTB..
3, As there are many risk factors for PTB or LBW, so it is better to adjust other confounders in the regression analysis if possible, for example, food consumption (especially the fish consumption) of pregnant women, the SES information of sample family, and the village dummy, etc.
Reply: Binary logistic regression indicate a relationship between LBW and low hair Hg exposure (OR: 3.34 95%CI: 1.17-9.58, p=0.025). While comparing low hair Hg exposure vs. medium/high hair Hg exposure to birthweight we found low Hg exposure to be associated with LBW (p=0.019). Multivariate logistic regression revealed that participants in the low tertile had significant higher odds of LBW compared with participants in the medium/high tertile hair Hg exposure (OR=5.3; 95% CI 1.3-20.6; p=0.016); this association was independent of their age, ethnic background and household income. We have added this to the result section lines 259-266.
Maternal fish consumption will be a topic of a separate paper that is in preparation.
4, If the fish consumption of pregnant women is available, it will be interesting to test the mediation effect of Hg concentration of pregnant woman on the association between fish consumption of pregnant women and birth weight (LWB or PTB).
Reply: As this is indeed a very interesting topic we are currently collecting and processing the data, so these will become available shortly. Results will be reported in a separate paper with additional nutritional information that was collected as well.
5, As gender is an important issue, it will be interesting to report the difference between girls and boys in the analysis.
Reply: thank you for this excellent suggestion. Based on guidance from our co-author, Dr. Emily Harville, a perinatal epidemiologist, we did a gender comparison. Overall there were more male infants (58%) born than female (42%). There was a higher incidence of PTB among male newborns, although not significant: (p=0.19; OR=1.98, 95% CI 0.81-5.07 for being a premature male child). We have added this to the result section lines 311-313.
LBW was more often seen in female newborns, but also not statistically significant (p=0.24; OR=0.49, 95% CI 0.17-1.48 for being a low birthweight male child). We have added this to the result section lines 314-315.
We also assessed the association between maternal hair-Hg concentrations and infant gender: there was no significant difference in maternal hair concentrations between female and male children (Mann-Whitney U = 3737, p=0.83). We have added this to the result section lines 270-272.
For the association between maternal age at birth and gender there was no difference in maternal age between female and male children (Mann-Whitney U = 4470, p=0.71). We have added this to the result section lines 316-317.
Round 2
Reviewer 2 Report
1,If the study aims to understand the association between prenatal mercury exposure and birth outcomes, it will be better to use all sample pregnant women from Suriname.
2, In the regression analysis, need to add the village dummy as independant variables, and the standard errors should be clustered at village level.
Author Response
Dear Reviewer,
Thank you for your valuable comments. we have tried to address them to the extent possible.
Comments and Suggestions for Authors
1,If the study aims to understand the association between prenatal mercury exposure and birth outcomes, it will be better to use all sample pregnant women from Suriname.
Reply: thank you for this comment. As indicated in the manuscript, our results show that the odds of having a low birthweight infant were inversely related to hair mercury concentrations. The lower the hair mercury concentration the higher the odds of a low birthweight infant. The distribution of mercury exposure in the three regions within Suriname where participants were recruited was almost non-overlapping – substantially higher in the interior compared to the other two regions. See table below. Hence to our opinion, the results from this interior sub-cohort merit a stand-alone publication. We have added this material to the introduction section lines 80-81 and results section lines 189-195.
|
|
Paramaribo |
Nickerie |
Interior |
|
N |
482 |
188 |
178 |
|
Range |
0.00-11.88 |
0.00-14.01 |
0.38-31.91 |
|
Mean +- SD |
0.81 +- 0.80 |
0.97 +- 1.18 |
5.65 +- 4.82 |
|
Median [IQR] |
0.63 [0.36 - 1.09] |
0.77 [0.50 -1.05] |
3.48 [2.01 - 9.01] |
2, In the regression analysis, need to add the village dummy as independent variables, and the standard errors should be clustered at village level.
Reply: thank you for this suggestion. We added the village dummy as an independent variable. We categorized the villages into downstream, upstream and none. The association between hair hg (low vs. medium/high) and LBW was also adjusted for the village location (downstream, upstream and none). This association remained significant.
|
Multivariate model for LBW |
|||||||||
|
|
B |
S.E. |
Wald |
df |
Sig. |
Exp(B) |
95% C.I.for EXP(B) |
||
|
Lower |
Upper |
||||||||
|
Step 1a |
hairhg low vs mediumhigh exposure(1) |
1.979 |
0.812 |
5.939 |
1 |
0.015 |
7.238 |
1.473 |
35.563 |
|
indigenous vs maroon |
0.974 |
0.863 |
1.275 |
1 |
0.259 |
2.649 |
0.488 |
14.368 |
|
|
household income <800 vs. 800+ SRD |
0.044 |
0.613 |
0.005 |
1 |
0.942 |
1.045 |
0.315 |
3.474 |
|
|
maternal age at intake 3-categories |
|
|
3.382 |
2 |
0.184 |
|
|
|
|
|
16-19 vs. 20-34 |
0.627 |
0.679 |
0.852 |
1 |
0.356 |
1.871 |
0.495 |
7.081 |
|
|
35+ vs. 20-34 |
1.302 |
0.724 |
3.230 |
1 |
0.072 |
3.676 |
0.889 |
15.207 |
|
|
village3 |
|
|
0.829 |
2 |
0.661 |
|
|
|
|
|
downstream vs. none |
0.988 |
1.465 |
0.455 |
1 |
0.500 |
2.687 |
0.152 |
47.414 |
|
|
upstream vs. none |
0.973 |
1.082 |
0.810 |
1 |
0.368 |
2.646 |
0.318 |
22.042 |
|
|
Constant |
-4.766 |
1.616 |
8.697 |
1 |
0.003 |
0.009 |
|
|
|
|
a. Variable(s) entered on step 1: hairhg low vs mediumhigh exposure, ethnicity 2 subgroups, household income 800 SRD, maternal age at intake 3-categories, village3. |
|||||||||
Multivariate logistic regression revealed that participants with low Hg exposure had significant higher odds of LBW compared with participants with medium/high hair Hg exposure (OR=7.2; 95% CI 1.5-35.6; p=0.015); this association was independent of their age, ethnic background, household income and village location. This information has been added to the abstract lines 27-28, method section lines 113-114, and results section lines 255-257